# The discharge process for a child with complex healthcare needs dependent on respiratory technology: A scoping review protocol

Cora O'Leary [ID]*, Rachel Flynn⁰, Serena FitzGerald, Helen Mulcahy [ID]⁰, Margaret Curtin⁰

School of Nursing and Midwifery, University College Cork, Cork, Ireland

⊗ These authors contributed equally to this work
* coraoleary@ucc.ie

## Abstract

### Background

Children with complex healthcare needs spend a long time in hospital with frequent delays in discharge. This may be due to fluctuating health status, or contextual factors such as training carers, social circumstances, suitable housing, and community resources such as homecare nurses. Understanding the process for an effective discharge for children with complex healthcare needs who are respiratory technology dependent is essential to guide clinical practice, and shape policy that supports timely, coordinated, and family-centered transitions from hospital to home and inform evidenced-based research.

### Objective

The aim of this scoping review is to gain an understanding of the discharge process for children with complex healthcare needs who are dependent on respiratory technology.

### Methods

CINAHL, Medline, PsycINFO and Embase databases will be used for the literature search. Articles will be considered from peer reviews sources and English language text from 2015 owing to the increase in children with complex healthcare needs requiring respiratory technology and advances in complex care. Title and abstracts followed by full-text articles will be reviewed to determine which papers meet the inclusion criteria. Two reviewers will review each stage with further discussion as required for exclusion and inclusion. Data extracted will include information on key findings related to the aim and objectives of the scoping review and will be summarized and presented in tables and charts with a narrative summary.

**Data availability statement:** No datasets were generated or analysed during the current study. All relevant data from this study will be made available upon study completion.

**Funding:** The author(s) received no specific funding for this work.

**Competing interests:** The authors have declared that no competing interests exist.

## Expected Results

International guidelines on care closer to home and caring for children with tracheostomies state the need for discharge pathways. This review will synthesis existing literature regarding the discharge processes of children with complex healthcare needs dependent on respiratory technology. Evidence will be organised across micro-level (child and family), meso-level (healthcare team and institutional processes), and macro-level (policy and system) influences. This scoping review will identify the multifaceted barriers and facilitator that impact the transition from hospital to home for children dependent on respiratory technology.

## Conclusion

This review will identify current evidence from relevant studies in relation to discharge processes and provide detail on components and context.

## Introduction

A child with complex healthcare needs is considered to have one or more complex chronic condition, substantial healthcare needs, functional limitations, and significant healthcare usage [1]. A child dependent on medical technology requires a device or devices to provide a vital body function, and can include tracheostomies, invasive or non-invasive ventilation and oxygen [2]. There continues to be international inconsistences in the terminology used to describe children with complex healthcare needs, resulting in reported prevalence rates ranging from 0.2% − 11.4% [3].

Children with complex healthcare needs often experience prolonged hospital stays, with discharge delays frequently driven not only by clinical factors, such as unstable ventilation changes, acute episodes of illness, but also by contextual factors such as caregiver availability, social circumstances, housing challenges, and limited community resources [4–6]. A delay in discharge occurs when a patient is deemed medically fit to be discharged but due to operational or logistical issues, they remain in hospital [6,7]. International evidence highlights significant delays in the discharge of children with complex healthcare needs who are dependent on respiratory technology []. Across four US hospital centres, discharge delays were observed in 68.5% (n = 37) of children with the average length of stay of 53.9 days (range from 4–204 days) [4]. Most of these cases (n = 31) involved children with tracheostomy tubes. Evidence from the UK and Ireland demonstrates significant discharge delays for children with complex healthcare needs. In a UK tertiary centre, delays of over 3 months were noted in 50% (n = 8) of admissions [8], while an Irish study reported average hospitalisations of 8.7 months (range 3–22 months) despite medial stability [9].

Prolonged stays in hospital for children with long-term health problems are at a greater risk of development delays due to lack of age-appropriate stimulation and by lack of social interaction and gaining age-appropriate skills [5,10,11]. It can affect parent child attachment and bonding, and due to sleep disturbances in a busy acute setting, learning, memory, and emotional regulation patterns can be affected [10].

Delays in discharge has been identified as a health disparity due to the detrimental effect on their development [5] and it can impact the whole family, such as time off work and alternative childcare arrangements for siblings, significant travel to care centres [12].

Although discharge processes for children with medical complexity have been described in narrative or single-centre studies, no comprehensive synthesis of factors influencing discharge for respiratory technology-dependent children exists. The scoping review therefore aims to explore the current discharge process for children with complex healthcare needs who are dependent on respiratory technology. This review will provide the necessary foundation for future research aimed at developing a discharge pathway for children dependent on respiratory technology. This pathway seeks to minimize hospital stays and align with the national framework for children with complex healthcare needs.

### Review question

What is known about the discharge processes for children with complex healthcare needs who are dependent on respiratory technology?

The objectives are:

1. What are the existing pathways for children with complex healthcare needs dependent on respiratory technology transitioning from hospital to home?

2. What factors impede the transition from hospital to home for children with complex healthcare needs dependent on respiratory technology?

3. What factors facilitate the discharge from hospital to home for children with complex healthcare needs dependent on respiratory technology?

4. What are the characteristics of an effective discharge for children with complex healthcare needs dependent on respiratory technology returning home?

## Methods

This review will be conducted using the guidelines by Joanna Briggs Institute (JBI) methodology for scoping reviews [13] and will be reported using Preferred Reporting Items for Systematic review and Meta-Analysis – Scoping reviews (PRISMA-ScR) reporting guidelines [14]. The protocol is being reported according to the items of the Preferred reporting Items for Systematic review and Meta-Analysis Protocols (PRISMA-P) checklist [15] (S1 File). The screening of records is scheduled for completion by November 2025, followed by data extraction by February 2026, will full completion of the scoping review expected by April 2026.

Ethical approval has not been sought for the present study because it is a review of published academic literature.

### Study eligibility criteria

Eligibility criteria are presented in Table 1 using the population, concept and context framework [17].

### Search strategy

Search terms were systematically identified through a rigorous process, which highlighted a total of 43 different terms used internationally for children with complex healthcare needs. Terms such as 'children with complex healthcare needs', 'children with medical complexity' and 'children with continuing care needs' searched with 'OR'. Additional searches related to discharge and respiratory technology will also be completed, combined with 'AND'. Major subject heading will be used in all databases. The databases which will be searched are CINAHL, Medline, PsycINFO and Embase. The

**Table 1. Eligibility criteria for study selection organised by population, exposure and outcome (PEO).**

| Population | Concept | Context | Additional eligibility criteria |
|---|---|---|---|
| Inclusion Criteria | Inclusion Criteria | Inclusion Criteria | Inclusion Criteria |
| Children < 18 years of age. Children with complex healthcare needs dependent on respiratory technology | 1. Studies examining the discharge process for children dependent on respiratory technology. 2. Studies reporting process related or clinical outcomes at the micro (patient), meso (healthcare setting), or macro (policy/societal) levels [16] | 1. Discharge pathways from hospital to home, including intermediate care settings 2. Consider clarifying whether discharges from community or home care settings to home are also included or excluded, as this is currently ambiguous. 3. Peer reviewed primary research from any geographical setting. | Peer reviewed primary research including qualitative and quantitative studies, mixed method, and descriptive papers will be included |
| Exclusion Criteria | Exclusion Criteria | Exclusion Criteria | Exclusion Criteria |
| Children with complex healthcare needs not dependent on respiratory technology | 1. Studies not addressing discharge processes or related clinical outcomes 2. Studies focusing on subsequent admissions or discharges after the initial discharge home | Subsequent admissions/ discharges after initial discharge home | Opinions pieces or letters to editors, articles that are not peer reviewed, will not be included in the scoping review. |

search strategy for the CINAHL database including the four main search terms in either title or abstract, MeSH subject headings, Boolean operators and study limitations applied are fully detailed and available in supplemental 2.

The reference lists of relevant reviews will be screened for additional relevant primary research studies. Only studies published in English and published between 2015–2025 will be considered. The timeframe chosen reflects advances in nursing and medical care, pharmacological and medical technology, and greater life expectancy for this cohort of children [1].

## Study/ Source of evidence Selection

All articles will be uploaded to Covidence, a software tool, used for conducting reviews [17]. It will be used to remove duplicates and for title/ abstract screening, full-text screening, data abstraction and quality assessment. If full articles are not available through databases, the corresponding author will be contacted for any missing full texts or data.

Five reviewers will be involved throughout all stages of screening, and all stages will be completed by two independent reviewers to enhance accuracy and reduce bias [17]. Titles/ abstracts will be screened independently by two reviewers. During this phase, each reviewer will independently assess the titles and abstracts of the articles identified in the search. Reviewers will classify each record as 'yes' (meets inclusion criteria), 'maybe' (uncertain or requires further discussion) or 'no' (does not met inclusion criteria). Any discrepancies will be reviewed and resolved through a discussion with a third reviewer at each stage of screening and data extraction. A PRISMA ScR flow chart [14] will be included to illustrate the study selection process, including reasons for exclusion and results from each stage of screening

## Data extraction

Data will be extracted from each included article, using a standardised data extraction tool in Covidence for evidence synthesis [17]. Data will be charted descriptively by title, authors, year of publication, country of publication and or where research was conducted, aims/ purpose, population and sample size, methodology, setting, discharge pathway, timeframes of discharge process, process and outcomes measures, contextual factors that hinder and facilitate discharges, and key findings in relation to scoping review questions.

The data extraction form will be piloted on a small number of articles to ensure all relevant data is identified early and ensure the table is logical [17]. During the pilot phase and data extraction phase, the form will be refined as needed. Any

modifications to the data extraction form will be documented and reported in the final scoping review. If additional information is required, this will be requested through the authors of the article.

## Data Analysis and presentation

Extracted data will be systematically reviewed and synthesized to address the aims of the scoping review. Data will be thematically categorized according to micro (patient-level), meso (institutional) and macro (policy/ system) factors. Information related to the research questions will be summarized using tables and thematic charts consistent with JBI guidelines [13], completed with a narrative synthesis. A narrative synthesis allows reviewers to compare studies, examine patterns and relationships in the data, and evaluate the robustness of the evidence [18]. All extracted data and synthesis tables will be made available in Cork Open Research Repository (CORA), the authors university of employment, which supports the deposit of the final manuscript version of publications for open access publication.

## Discussion

This scoping review protocol outlines a comprehensive synthesis of peer-reviewed literature examining the initial hospital-to-home discharge process for children with complex healthcare needs who are dependent on respiratory technology, with the aim of identifying critical gaps, challenges, and opportunities for improvement. The factors identified through the review will be used for a proposed discharge pathway for children with complex healthcare needs dependent on respiratory technology.

One of the limitations of this review is the exclusion of grey literature and whilst there is grey literature published on this topic, the authors chose to focus this review on peer-reviewed sources. Four databases will be searched, which may result in articles in other databases or not correctly indexed articles being missed. A further limitation of this study is the inclusion only of English-language literature which may introduce geographical and linguistic bias.

Findings from this scoping review will be disseminated through peer-reviewed journal, national and international conferences, and both formal and informal teaching platforms to reach key stakeholders, policymakers, and clinicians. Findings from the scoping review will inform the next stage of research which will focus on the development of a discharge pathway. This will be undertaken in partnership with patient and public involvement stakeholders situated within the context of national policy priorities on integrated and delivering care closer to home.

## Supporting information

**S1 File. PRISMA Checklist.**
(DOC)

**S2 File. CINAHL Search Strategy.**
(DOCX)

## Acknowledgments

No acknowledgements to disclose.

## Author contributions

**Conceptualization:** Cora O#39;Leary.

**Methodology:** Cora O#39;Leary.

**Writing – original draft:** Cora O#39;Leary.

**Writing – review & editing:** Cora O#39;Leary, Serena FitzGerald, Margaret Curtin, Rachel Flynn, Helen Mulcahy.

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
