## [Decision Letter · Decision Letter 0]

3 Nov 2025

PONE-D-25-53762The discharge process for a child with complex healthcare needs dependent on respiratory technology: A scoping review protocolPLOS ONE

Dear Dr. O'Leary,

Thank you for submitting your manuscript to PLOS ONE. After careful consideration, we feel that it has merit but does not fully meet PLOS ONE’s publication criteria as it currently stands. Therefore, we invite you to submit a revised version of the manuscript that addresses the points raised during the review process.

We look forward to receiving your revised manuscript.

Kind regards,

André Luis C Ramalho, PhD

Academic Editor

PLOS ONE

Journal Requirements:

2. Your abstract cannot contain citations. Please only include citations in the body text of the manuscript, and ensure that they remain in ascending numerical order on first mention.

**Additional Editor Comments:**

Dear Dr. O’Leary and colleagues,

Your manuscript addresses a highly relevant and underexplored topic in pediatric complex care. The proposed scoping review is well-conceived, methodologically grounded, and has clear potential to inform both clinical practice and health policy on discharge processes for technology-dependent children. The writing demonstrates strong familiarity with current literature and appropriate use of recognized frameworks such as JBI and PRISMA-ScR. Following evaluation of the two independent peer reviews and an additional editorial assessment, I have decided that the manuscript requires a Major Revision before it can be considered for publication. Both reviewers provided thoughtful and constructive feedback, identifying complementary aspects for improvement, and the editorial review has also identified several additional issues not fully addressed in their comments.

Summary of Required Revisions

In line with the reviewers’ observations:

- Reformat the abstract into a structured version (Background, Objectives, Methods, Expected Results, Conclusion).

- Replace the PEO framework with the PCC (Population–Concept–Context) format, in accordance with JBI guidance.

- Correct the numbering of objectives, and clearly define “effective discharge” and “respiratory technology.”

- Include examples of search terms and Boolean logic, justify the 2015 start date, and specify how data will be organized and analyzed (software, thematic framework).

- Provide a concrete data availability plan (e.g., Zenodo, OSF).

- Add a brief ethics statement confirming that no human or identifiable data are involved.

- Expand the discussion to explain how results will inform practice, pathway design, and policy, and briefly mention dissemination plans.

- Review the references and language for consistency, grammar, and DOIs.

Additional points identified by the editorial review:

- The introduction, although comprehensive, could be more focused on the knowledge gap justifying this review, avoiding excessive repetition of descriptive data.

- Consider registering the protocol in PROSPERO or the Open Science Framework (OSF) to enhance transparency.

- Clarify how quality assurance and inter-reviewer agreement will be monitored during screening and data extraction.

- The manuscript would benefit from shorter paragraphs and clearer transitions to improve readability.

- Ensure terminology consistency across all sections—use “children with complex healthcare needs” uniformly.

- The statement of aims and policy relevance could be strengthened by explicitly linking expected outputs to clinical and system-level decision-making.

Once these revisions are made, the manuscript will likely meet PLOS ONE’s standards for methodological transparency and scientific clarity. The topic is of clear international importance, and the proposed protocol will serve as a valuable foundation for evidence synthesis and future pathway development in pediatric complex care.

Please submit a revised manuscript that thoroughly addresses, point by point, all comments raised by the reviewers and the editorial team.

Best regards

Reviewers' comments:

Reviewer's Responses to Questions

**Comments to the Author**

1. Does the manuscript provide a valid rationale for the proposed study, with clearly identified and justified research questions?

Reviewer #1: Yes

Reviewer #2: Yes

2. Is the protocol technically sound and planned in a manner that will lead to a meaningful outcome and allow testing the stated hypotheses?

Reviewer #1: Yes

Reviewer #2: Partly

3. Is the methodology feasible and described in sufficient detail to allow the work to be replicable?

Reviewer #1: Yes

Reviewer #2: Yes

4. Have the authors described where all data underlying the findings will be made available when the study is complete?

Reviewer #1: No

Reviewer #2: No

5. Is the manuscript presented in an intelligible fashion and written in standard English?

Reviewer #1: Yes

Reviewer #2: Yes

6. Review Comments to the Author

You may also provide optional suggestions and comments to authors that they might find helpful in planning their study.

Reviewer #1: Thank you for the opportunity to review your protocol titled “The discharge process for a child with complex healthcare needs dependent on respiratory technology.” This is a thoughtful and well-prepared manuscript that addresses a highly relevant and underexplored area in pediatric complex care. The topic is timely, and the proposed review has the potential to make an important contribution to improving discharge planning and care coordination for this population.

Overall, the paper is clearly written, logically structured, and follows recognized frameworks such as JBI and PRISMA-ScR. The rationale for the review is compelling, and the objectives are clearly aligned with the main research question. I particularly appreciate the comprehensive background section, which captures both the clinical and contextual factors influencing discharge delays.

Before acceptance, I would suggest a few minor revisions to strengthen the protocol:

Data Availability:

Please include a brief statement on where the data from this review will be made available once completed. For example, you could note whether the extracted data and summary tables will be shared in a public repository or made available upon reasonable request.

Objective 5 Missing:

In the “Objectives” section, numbering stops at point 4, and Objective 5 is left blank. Please review and clarify whether an additional objective was intended or if this was a formatting oversight.

Clarify the Review Question:

The main question could be phrased more in line with the exploratory purpose of a scoping review, for example, “What is known about the discharge processes for children with complex healthcare needs who are dependent on respiratory technology?”

Framework Alignment:

The inclusion criteria are currently presented using the Population–Exposure–Outcome (PEO) format. Scoping reviews usually use the Population–Concept–Context (PCC) framework recommended by JBI. Adjusting this would bring your protocol fully in line with best practice.

Ethical Transparency:

Since this is a scoping review of published studies, please include a short statement confirming that ethical approval is not required and that no individual-level or identifiable data will be used.

Terminology and Consistency:

Terms like “children with complex healthcare needs” and “children with medical complexity” are used interchangeably. It may help to define your preferred term early in the introduction and use it consistently throughout.

Minor Editing Suggestions:

Break up a few long paragraphs for readability.

Make sure all abbreviations (such as PRISMA-ScR and JBI) are spelled out the first time they appear.

Reviewer #2: This manuscript is a scoping review protocol which aims to analyse evidence on the discharge process for children with complex healthcare needs (CCHNs) and being dependent on respiratory technology. The topic is highly relevant and timely, addressing a growing population of children requiring complex discharge planning, and fits well within PLOS ONE’s scope of methodological rigour and healthcare systems research.

The title is clear, descriptive, and aligns well with PLOS ONE’s protocol standards The abstract reads more like a general summary than a structured scientific abstract. It lacks subheadings (Objective, Introduction, Methods, etc.), which are preferred for protocols. Key methodological parameters (e.g., inclusion/exclusion rationale, expected number of reviewers, data charting tools, timeline) are mentioned but not sufficiently detailed. The inclusion of the word “protocol” in the short title is appropriate but could be accompanied by “scoping review protocol following JBI and PRISMA-ScR” for indexing precision. I would suggest that you reformat the abstract with clear subheadings (Background, Objectives, Methods, Expected Results, Conclusion), and clarify the timeframe (“since June 2015”) and justify it (e.g., relevance to advances in complex care or technology-dependent paediatrics).

The introduction is overly descriptive and literature-heavy, sometimes blurring the line between background and results of prior reviews. Some references (e.g., U.S. and Irish studies on discharge delays) are cited without explicit interpretation or critical comparison. The problem statement could be sharper in identifying the gap in existing evidence synthesis (e.g., absence of scoping reviews mapping this process). There are minor inconsistencies in referencing format (e.g., punctuation, spacing around years and semicolons). What you should do is to condense background data (lines 83–150) to focus on the rationale for conducting a scoping rather than a systematic review. Explicitly state the knowledge gap, for instance: “Although discharge processes for children with medical complexity have been described in narrative or single-centre studies, no comprehensive synthesis of factors influencing discharge for respiratory technology-dependent children exists.” Link the introduction directly to how this protocol addresses practice or policy gaps (e.g., informing discharge pathway design, as mentioned on line 155).

Objectives could be mapped to the PCC (Population–Concept–Context) structure for alignment with JBI guidance. The last objective (“What constitutes an effective discharge…”) should specify the operational definition of “effective”, e.g., timely, safe, family-centred. Present the objectives using PCC format. Define “respiratory technology” (invasive/non-invasive ventilation, tracheostomy, long-term oxygen) early in the objectives.

As far as the search strategy is concerned: “Search terms were systematically identified” (line 190), but lack any example of search strings or keywords. They should at least summarise the key concepts (children, complex healthcare needs, respiratory technology, discharge, transition). Supplement S2 is cited but not briefly described in the text. Date restriction (“since June 2015”) is arbitrary, and its justification is missing. Language restriction to English may exclude relevant studies from non-English European countries. Exclusion of grey literature (line 245) is a notable limitation given that discharge process data often arise from policy documents, hospital reports, or theses. There is no stakeholder consultation described, which JBI recommends as an optional but valuable step. In the data charting form content is listed (line 216–221) but not illustrated. In the analysis “Narrative synthesis” (line 234) is appropriate but should specify which analytical approach will be applied (e.g., thematic synthesis, mapping by micro/meso/macro levels). Ethics, although correctly stated as not required, could be briefly mentioned in the Methods for completeness. My advice is to summarise the search strategy within the text and include Boolean logic example (“complex healthcare needs” AND “respiratory technology” AND “discharge”). In addition, justify the 2015 date cutoff (e.g., shift in paediatric home ventilation policies). Consider including grey literature to enhance comprehensiveness; if excluded, provide stronger justification. Add a short statement acknowledging the potential contribution of consultation with stakeholders (families, clinicians, discharge coordinators). Clarify analytic framework—e.g., “Data will be thematically categorized according to micro (patient-level), meso (institutional), and macro (policy/system) factors.”

In the Results section, there is no plan for handling missing or incomplete data described (e.g., contacting authors). It also lacks mention of software for managing extracted data (e.g., Excel, NVivo). The protocol omits a plan for charting the number of studies by design, country, and year, which is expected in scoping reviews. Include statement: “Data will be charted descriptively by study type, country, setting, and design.” Specify that results will be summarised via tables and thematic charts (consistent with JBI format). Add a brief plan for quality assurance, even if critical appraisal is not required for scoping reviews.

The discussion (lines 238–249) is brief and reads more like a reiteration of the purpose than an analytical reflection. It could elaborate on how findings will inform practice (e.g., hospital discharge protocols) and policy (e.g., national complex care strategies). It does not mention a dissemination plan beyond publication (“will be published in a peer-reviewed journal”). I recommend that you expand the discussion to clarify anticipated contributions to practice, research, and policy. Also, suggest inclusion of how the resulting map will support families or inform future interventions. You can add a note on planned dissemination (e.g., conference presentation, stakeholder summary).

There is inconsistent formatting in the references (e.g., some missing periods or irregular capitalisation; Pediatric Otolaryngology vs Journal of Pediatrics). Some DOIs are missing, though PLOS requires them. References 12–15 include mixed styles (numbering before sentence punctuation). Please revise references to conform strictly to PLOS ONE’s numeric citation style with DOIs for all journal articles. Consider reducing redundancy (some overlapping citations for delayed discharge).

There are several grammatical issues and inconsistent tense (e.g., “The inclusion criteria is presented…” → should be “are presented”). There is an overuse of long sentences, which reduces clarity. There is inconsistent use of “children with complex healthcare needs” vs. “children with medical complexity.” Please amend these by proofreading for grammar and tense consistency, and define and consistently use one term (“children with complex healthcare needs”). Use concise, active voice throughout (e.g., “This review will identify…” rather than “It is expected that this review will identify…”).

The “Data Availability” statement (page 5) repeats standard text without specifying a repository plan. Include a short forward-looking statement: “All extracted data and synthesised tables will be made available in an open-access repository (e.g., Zenodo, Open Science Framework) upon publication.”

7. PLOS authors have the option to publish the peer review history of their article (what does this mean?). If published, this will include your full peer review and any attached files.

Reviewer #1: **Yes:** Fatima Abubakar Ishaq

Reviewer #2: No

---

## [Author Response · Author response to Decision Letter 1]

18 Dec 2025

We would like to thank the editor and the reviewers for their comments. All comments were taken into consideration and tracked on the document in files portal

---

## [Decision Letter · Decision Letter 1]

1 Feb 2026

PONE-D-25-53762R1The discharge process for a child with complex healthcare needs dependent on respiratory technology: A scoping review protocolPLOS One

Dear Dr. O'Leary,

Thank you for submitting your manuscript to PLOS ONE. After careful consideration, we feel that it has merit but does not fully meet PLOS ONE’s publication criteria as it currently stands. Therefore, we invite you to submit a revised version of the manuscript that addresses the points raised during the review process.

We look forward to receiving your revised manuscript.

Kind regards,

Maher Abdelraheim Titi

Academic Editor

PLOS One

Journal Requirements:

Additional Editor Comments:

Thank you for your revised submission. Upon review, it appears that Reviewer 2’s comments were not fully addressed in your latest response. To proceed toward acceptance, please ensure that all reviewer feedback is explicitly responded to and incorporated into the manuscript where appropriate.

In addition, the following points require further clarification and revision:

• Please provide a clearer and more robust rationale for excluding grey literature.

• Clarify how disagreements during data extraction will be resolved.

• Specify whether the narrative synthesis will follow a recognized framework.

Reviewer 2 Comments:

This manuscript is a scoping review protocol which aims to analyse evidence on the discharge process for children with complex healthcare needs (CCHNs) and being dependent on respiratory technology. The topic is highly relevant and timely, addressing a growing population of children requiring complex discharge planning, and fits well within PLOS ONE’s scope of methodological rigour and healthcare systems research.

The title is clear, descriptive, and aligns well with PLOS ONE’s protocol standards The abstract reads more like a general summary than a structured scientific abstract. It lacks subheadings (Objective, Introduction, Methods, etc.), which are preferred for protocols. Key methodological parameters (e.g., inclusion/exclusion rationale, expected number of reviewers, data charting tools, timeline) are mentioned but not sufficiently detailed. The inclusion of the word “protocol” in the short title is appropriate but could be accompanied by “scoping review protocol following JBI and PRISMA-ScR” for indexing precision. I would suggest that you reformat the abstract with clear subheadings (Background, Objectives, Methods, Expected Results, Conclusion), and clarify the timeframe (“since June 2015”) and justify it (e.g., relevance to advances in complex care or technology-dependent paediatrics).

The introduction is overly descriptive and literature-heavy, sometimes blurring the line between background and results of prior reviews. Some references (e.g., U.S. and Irish studies on discharge delays) are cited without explicit interpretation or critical comparison. The problem statement could be sharper in identifying the gap in existing evidence synthesis (e.g., absence of scoping reviews mapping this process). There are minor inconsistencies in referencing format (e.g., punctuation, spacing around years and semicolons). What you should do is to condense background data (lines 83–150) to focus on the rationale for conducting a scoping rather than a systematic review. Explicitly state the knowledge gap, for instance: “Although discharge processes for children with medical complexity have been described in narrative or single-centre studies, no comprehensive synthesis of factors influencing discharge for respiratory technology-dependent children exists.” Link the introduction directly to how this protocol addresses practice or policy gaps (e.g., informing discharge pathway design, as mentioned on line 155).

Objectives could be mapped to the PCC (Population–Concept–Context) structure for alignment with JBI guidance. The last objective (“What constitutes an effective discharge…”) should specify the operational definition of “effective”, e.g., timely, safe, family-centred. Present the objectives using PCC format. Define “respiratory technology” (invasive/non-invasive ventilation, tracheostomy, long-term oxygen) early in the objectives.

As far as the search strategy is concerned: “Search terms were systematically identified” (line 190), but lack any example of search strings or keywords. They should at least summarise the key concepts (children, complex healthcare needs, respiratory technology, discharge, transition). Supplement S2 is cited but not briefly described in the text. Date restriction (“since June 2015”) is arbitrary, and its justification is missing. Language restriction to English may exclude relevant studies from non-English European countries. Exclusion of grey literature (line 245) is a notable limitation given that discharge process data often arise from policy documents, hospital reports, or theses. There is no stakeholder consultation described, which JBI recommends as an optional but valuable step. In the data charting form content is listed (line 216–221) but not illustrated. In the analysis “Narrative synthesis” (line 234) is appropriate but should specify which analytical approach will be applied (e.g., thematic synthesis, mapping by micro/meso/macro levels). Ethics, although correctly stated as not required, could be briefly mentioned in the Methods for completeness. My advice is to summarise the search strategy within the text and include Boolean logic example (“complex healthcare needs” AND “respiratory technology” AND “discharge”). In addition, justify the 2015 date cutoff (e.g., shift in paediatric home ventilation policies). Consider including grey literature to enhance comprehensiveness; if excluded, provide stronger justification. Add a short statement acknowledging the potential contribution of consultation with stakeholders (families, clinicians, discharge coordinators). Clarify analytic framework—e.g., “Data will be thematically categorized according to micro (patient-level), meso (institutional), and macro (policy/system) factors.”

In the Results section, there is no plan for handling missing or incomplete data described (e.g., contacting authors). It also lacks mention of software for managing extracted data (e.g., Excel, NVivo). The protocol omits a plan for charting the number of studies by design, country, and year, which is expected in scoping reviews. Include statement: “Data will be charted descriptively by study type, country, setting, and design.” Specify that results will be summarised via tables and thematic charts (consistent with JBI format). Add a brief plan for quality assurance, even if critical appraisal is not required for scoping reviews.

The discussion (lines 238–249) is brief and reads more like a reiteration of the purpose than an analytical reflection. It could elaborate on how findings will inform practice (e.g., hospital discharge protocols) and policy (e.g., national complex care strategies). It does not mention a dissemination plan beyond publication (“will be published in a peer-reviewed journal”). I recommend that you expand the discussion to clarify anticipated contributions to practice, research, and policy. Also, suggest inclusion of how the resulting map will support families or inform future interventions. You can add a note on planned dissemination (e.g., conference presentation, stakeholder summary).

There is inconsistent formatting in the references (e.g., some missing periods or irregular capitalisation; Pediatric Otolaryngology vs Journal of Pediatrics). Some DOIs are missing, though PLOS requires them. References 12–15 include mixed styles (numbering before sentence punctuation). Please revise references to conform strictly to PLOS ONE’s numeric citation style with DOIs for all journal articles. Consider reducing redundancy (some overlapping citations for delayed discharge).

There are several grammatical issues and inconsistent tense (e.g., “The inclusion criteria is presented…” → should be “are presented”). There is an overuse of long sentences, which reduces clarity. There is inconsistent use of “children with complex healthcare needs” vs. “children with medical complexity.” Please amend these by proofreading for grammar and tense consistency, and define and consistently use one term (“children with complex healthcare needs”). Use concise, active voice throughout (e.g., “This review will identify…” rather than “It is expected that this review will identify…”).

The “Data Availability” statement (page 5) repeats standard text without specifying a repository plan. Include a short forward-looking statement: “All extracted data and synthesised tables will be made available in an open-access repository (e.g., Zenodo, Open Science Framework) upon publication.”

Reviewers' comments:

Reviewer's Responses to Questions

**Comments to the Author**

1. Does the manuscript provide a valid rationale for the proposed study, with clearly identified and justified research questions?

Reviewer #3: Yes

Reviewer #4: Yes

2. Is the protocol technically sound and planned in a manner that will lead to a meaningful outcome and allow testing the stated hypotheses?

Reviewer #3: Yes

Reviewer #4: Yes

3. Is the methodology feasible and described in sufficient detail to allow the work to be replicable?

Reviewer #3: Yes

Reviewer #4: Yes

4. Have the authors described where all data underlying the findings will be made available when the study is complete?

Reviewer #3: Yes

Reviewer #4: Yes

5. Is the manuscript presented in an intelligible fashion and written in standard English?

Reviewer #3: Yes

Reviewer #4: Yes

6. Review Comments to the Author

You may also provide optional suggestions and comments to authors that they might find helpful in planning their study.

Reviewer #3: No further recommendations, agree with all previous recommendations and subsequent changes. This is a very relevant study to the changing population needs in the community.

Reviewer #4: The revisions of the authors are appropriate and address reviewer concerns adequately and the article ramins one on a topic of importance.

7. PLOS authors have the option to publish the peer review history of their article (what does this mean?). If published, this will include your full peer review and any attached files.

Reviewer #3: **Yes:** Sophie Ronan

Reviewer #4: No

---

## [Author Response · Author response to Decision Letter 2]

18 Mar 2026

Dear Editor and Reviewer

Thank you for the detailed comments and recommendations. All the authors reviewed and updated manuscript as per the recommendations

Regards

---

## [Editor Report · Decision Letter 2]

29 Mar 2026

PONE-D-25-53762R2The discharge process for a child with complex healthcare needs dependent on respiratory technology: A scoping review protocol following JBI and PRISMA-ScR GuidelinesPLOS One

Dear Dr. O'Leary,

Thank you for submitting your manuscript to PLOS ONE. After careful consideration, we feel that it has merit but does not fully meet PLOS ONE’s publication criteria as it currently stands. Therefore, we invite you to submit a revised version of the manuscript that addresses the points raised during the review process.

The manuscript is much improved; however, further refinement is still needed. Strengthening the flow, improving coherence, and removing remaining redundancies would significantly enhance clarity and readability. My detailed comments below highlight areas where repetition persists and where restructuring could improve the overall presentation.

We look forward to receiving your revised manuscript.

Kind regards,

Maher Abdelraheim Titi

Academic Editor

PLOS One

Journal Requirements:

Additional Editor Comments:

1. Although Reviewer 2 recommended adding ‘A scoping review protocol following JBI and PRISMA‑ScR Guidelines’ as the short title, I recommend keeping the shorter version: ‘A scoping review protocol.

2. The statement ‘Data extracted will include information on key findings related to the aim and objectives of the scoping review’ can be moved to the following sentence, as the main focus of the preceding statement is on the review and screening procedures, followed by data extraction.”

3. It is not common practice to report the ethical statement in the abstract. According to the PLOS ONE guidelines, it should instead be included under the ‘Ethics Statement’ subheading.

4. Rephrase the expected results statements for better clarity and avoid redundancy consider the following ideas : what the review will produce, how the results will be organized( i.e., Evidence will be organised across micro‑level (child and family), meso‑level (healthcare team and institutional processes), and macro‑level (policy and system) influences., and what type of knowledge the review will generate.

5. Redundancy in terminology has been noted. For example, the term ‘healthcare needs’ is repeated within the same sentence in lines 71–73, and the word ‘technologies’ appears redundantly in lines 76–78. Please consider removing such repetition to improve readability and flow.

6. Consider rewriting the statement in lines 78–80 as follows to improve clarity:

“There continue to be international inconsistencies in the terminology used to describe children with complex healthcare needs, resulting in reported prevalence rates ranging from 0.2% to 11.4%.”

7. For line 86, consider rephrasing “A delay in discharge is when” to: “A delay in discharge occurs when a patient…”

8. For improved readability, consider rewriting the statements in lines 90–95 as follows:

“Across four US hospital centres, discharge delays were observed in 68.5% (n=37) of children, with an average length of stay of 53.9 days (range 4–204 days). Most of these cases (n=31) involved children with tracheostomy tubes.”

9. Line 94-101: Consider removing the notation (n=8 or 13 etc )’ or replacing it with a written expression to improve readability and flow.

10. In lines 103–104, the phrase ‘can impact on early childhood development and’ may be removed to avoid redundancy and improve clarity.

11. Lines 115–116: Consider rephrasing ‘The aim of this scoping review is to explore’ to ‘This scoping review therefore aims to…’ to improve flow and alignment with the preceding sentence.

12. Lines 117–120: This section would benefit from revision to more clearly emphasise the specific advantages for children who are dependent on respiratory technologies.

13. Lines 138 and 140: Delete the phrase ‘hereafter known as’; using the abbreviation (JBI) alone is sufficient.

14. The statement in line 142-144” rephrase “The screening of records is scheduled for completion by November 2025, followed by data extraction by February 2026, with full completion of the scoping review expected by April 2026.

15. Statement line 145-146: According to the PLOS ONE guidelines, it should be included under the ‘Ethics Statement’ subheading.

16. Table 1 lacks clarity, and the following points should be addressed to improve accuracy and alignment with the PCC framework:

a. Remove Redundancy: There is no need to repeat “Young people >18 years of age” in the exclusion criteria, as the age range is already clearly defined in the inclusion criteria.

b. Rephrase and Clarify Statements Under Inclusion Criteria – Concept:

1. Studies examining the discharge process for children dependent on respiratory technology.

2. Studies reporting process related or clinical outcomes at the micro (patient), meso (healthcare setting), or macro (policy/societal) levels

c. Rephrase and Clarify Statements Under Exclusion Criteria – Concept:

1. Studies not addressing discharge processes or related clinical outcomes

2. Studies focusing on subsequent admissions or discharges after the initial discharge home

d. Clarify Context Criteria: Under Inclusion Criteria:

1. Discharge pathways from hospital to home, including intermediate care settings

2. Consider clarifying whether discharges from community or home care settings to home are also included or excluded, as this is currently ambiguous.

3. Peer reviewed primary research from any geographical setting

e. Remove or Relocate Unnecessary Narrative Content

i. The following statement in Table 1 lacks clarity and appears out of place:

“Setting from hospital to home may include several intermediate settings. In a retrospective review in the US from 2016–2019, 22% of children were discharged to other facilities [19].”

This information is descriptive rather than a criterion and should be removed from the table.

f. The inclusion and exclusion criteria related to study type—such as opinion pieces, letters to the editor, non–peer‑reviewed publications, and language restrictions—can be added to Table 1 under a new column titled ‘Additional Eligibility Criteria.

17. The paragraph “Types of sources” (lines 152–156) can be removed if the authors decide to add an “Additional Eligibility Criteria” column to Table 1

18. Lines 178–179 should be rephrased as follows: “The corresponding author will be contacted for any missing full texts or data.” Consider the repetition in line 209-2010.

It is important to clearly state how missing information will be handled.

19. The statements in line 176-177 is duplicated with statement 197-198 related to using Covidence tool with 2 different references please reconcile and remove duplicate

20. The statement in lines 185–186 is duplicated in lines 189 regarding the involvement of a third reviewer. Please remove the duplication to ensure consistency and avoid repetition. The same issue appears with the statements in lines 181 and 192–194. I recommend reviewing the entire paragraph to improve readability and eliminate repetitive content.

21. Line 220: Please ensure that the abbreviation “CORA” is defined at first use. If it has already been defined earlier in the manuscript, avoid repeating the full term and use only the abbreviation.

22. Please add the language restriction as a study limitation. Limiting the review to English‑language studies should be explicitly acknowledged under the “Limitations” section

---

## [Author Response · Author response to Decision Letter 3]

21 Apr 2026

Dear Editor

Thank you for all the suggestions and comments. We reviewed each comment and made amendments as per the recommendations

Regards

---

## [Editor Report · Decision Letter 3]

5 May 2026

The discharge process for a child with complex healthcare needs dependent on respiratory technology: A scoping review protocol

PONE-D-25-53762R3

Dear Dr. O'Leary,

We’re pleased to inform you that your manuscript has been judged scientifically suitable for publication and will be formally accepted for publication once it meets all outstanding technical requirements.

Kind regards,

Maher Abdelraheim Titi

Academic Editor

PLOS One
---

## [Editor Report · Acceptance letter]

PONE-D-25-53762R3

PLOS One

Dear Dr. O'Leary,

I'm pleased to inform you that your manuscript has been deemed suitable for publication in PLOS One. Congratulations! Your manuscript is now being handed over to our production team.

Kind regards,

on behalf of

Dr. Maher Abdelraheim Titi

Academic Editor

PLOS One